# Household Income Diversification and Food Insecurity: A Case Study of Afghanistan

**DOI:** 10.3390/foods14122048

**Published:** 2025-06-10

**Authors:** Rohullah Niazi, Aijun Liu, Jiqin Han, Sherouk Hassan, Hamidullah Elham, Maurice Osewe

**Affiliations:** 1College of Economics and Management, Nanjing Agricultural University, Nanjing 210095, China; ruhullah.niazi@gmail.com (R.N.); jhan@njau.edu.cn (J.H.); sa1918@fayoum.edu.eg (S.H.); 2Department of Agricultural Economics and Extension, Faculty of Agriculture, Helmand University, Peace Watt, Lashkar Gah 3901, Helmand, Afghanistan; hamidullah.ilham@gmail.com; 3Jin Shanbao Institute for Agriculture and Rural Development, Nanjing Agricultural University, 1 Weigang, Nanjing 210095, China; 4China Center for Food Security Studies, Nanjing Agricultural University, Nanjing 210095, China; 5Agricultural Economics and Farm Surveys, Rural Economy & Development Programme, Teagasc Mellows Campus, Athenry, Co., H65 R718 Galway, Ireland; maurice.osewe@teagasc.ie

**Keywords:** Afghanistan, Helmand, food insecurity, income diversification

## Abstract

Strategies for income diversification are crucial in enhancing resilience to food insecurity, particularly in developing countries like Afghanistan, where resource scarcity and socio-political conflicts hinder economic stability. This study examines the relationship between income diversification and household food insecurity in Helmand, Afghanistan. Using cross-sectional survey data from 302 household heads, multivariate ordered logistic regression analysis was conducted to assess the association between income diversification and household food insecurity. Results indicate that households with lower income diversification are more likely to experience severe food insecurity. Additionally, factors such as illiteracy, lack of land ownership, unchanged income from the previous year, the absence of relatives in government jobs, poor financial situation, higher numbers of female household members, and greater distance from the district bazaar were associated with increased food insecurity. The findings suggest that while income diversification plays a vital role in mitigating food insecurity, it is insufficient alone for some households. The study concludes with recommendations to promote income diversification, improve education, facilitate land distribution, expand government employment programs, empower women, and improve market access and infrastructure in regions like Helmand to combat food insecurity.

## 1. Introduction

Addressing the nutritional needs of an expanding global population is a significant concern for governments and policymakers globally [1]. Even with continued global efforts to fight hunger, around 733 million people still did not have enough to eat in 2023, according to the FAO, showing that food insecurity remains a serious global challenge [1]. Mitigating food insecurity necessitates comprehensive interventions centered on four essential pillars: stability, access, utilization, and availability [2]. Insufficient food production is an issue, but price volatility, poverty, poor infrastructure, and weak economies are all key factors, particularly in vulnerable countries [3,4]. Climate change intensifies these issues by changing agricultural production, amplifying weather extremes, and destabilizing food systems, especially in areas with significant agricultural dependence and restricted adaptation ability, such as South Asia [5,6].

Diversifying income has become a crucial technique within the sustainable livelihood framework to improve food security [7,8]. Characterized as the diversification of income streams through many activities [9], it enhances household resilience to climate change, market volatility, and other disruptions [10]. Households having many income sources might diminish susceptibility by alleviating the risks linked to dependence on a singular revenue stream [11,12]. This advantage often benefits wealthier households with better access to financial resources and markets, enabling them to invest in more profitable industries [13]. Studies from Sub-Saharan Africa, South Asia, and Latin America demonstrate that households with diverse income sources are more resilient to economic shocks and food security challenges [6,14].

In Afghanistan, decades of conflict and environmental degradation contribute to food security issues, especially in rural areas [15,16]. With around 80 percent of the population engaged in agriculture, Afghanistan is particularly vulnerable to food insecurity, given its low productivity and susceptibility to climate fluctuations [17,18]. Limited infrastructure, such as poor transportation networks, unreliable electricity, and a lack of cold storage facilities, restricts market access, preventing rural households from selling agricultural products at competitive prices [19]. Furthermore, inadequate water management systems exacerbate the impact of climate change and erratic weather patterns, limiting agricultural productivity, particularly in highly water-dependent sectors like wheat and rice farming [20]. In addition, the absence of formal financial services, such as microcredit, insurance, and banking systems, further worsens the vulnerability of Afghan farmers, as they lack the means to invest in productivity-enhancing technologies or to smooth income variability from crop failures [21]. Income diversification, which typically includes agricultural and non-agricultural activities, has been a key strategy for Afghan households to cope with economic shocks [22]. However, the conflict not only disrupts agricultural activities but also increases the cost of input, makes transportation risky, and limits access to education and skills development, which could otherwise improve the capacity for diversification [16].

Afghanistan’s rural economy is particularly vulnerable to the adverse effects of environmental stressors, with agricultural productivity projected to decline by up to 20% by 2050 due to climate-related factors such as droughts and floods [23]. This makes diversification an even more vital strategy for mitigating food insecurity. However, the success of diversification efforts is limited by limited access to social capital, markets, and financial services, which reduces their effectiveness in improving food security in rural areas [16,24]. As a result, despite the potential benefits of income diversification, its overall impact on food security is still limited by these various challenges, which hinder the full effectiveness of diversification strategies in rural areas of Afghanistan. Along with environmental challenges, food geopolitics play a major role in food security, especially in areas affected by conflict. Things like trade policies, sanctions, and international supply chains directly impact the cost and availability of essential agricultural supplies like fertilizers, seeds, and machinery. In countries like Afghanistan, which rely heavily on imports for these resources, changes in global trade can make food insecurity worse, further destabilizing local food systems [6].

This study investigates the relationship between household income diversification and food insecurity in Afghanistan. By examining whether households with diversified income sources are better equipped to achieve food security among socio-economic challenges, this research contributes to a deeper understanding of how diversification strategies may alleviate food insecurity. The findings will provide valuable insights for policymakers and development practitioners working to enhance resilience and food security in conflict-affected regions.

### 1.1. Literature Review

Income diversification is essential for family resilience, providing various agricultural and non-agricultural activities that enable households to obtain diverse income sources and handle risks efficiently. Agricultural diversification encompasses activities like integrated farming, crop diversification, and raising livestock, practices, which are crucial for offering households varied agricultural options that serve as a buffer against market and environmental instability [25]. Conversely, non-agricultural or non-farm diversification encompasses manufacturing, service-oriented activities, and jobs in wages, that generally yield more stable and potentially greater returns, particularly in regions where agricultural profitability is limited [26]. Households frequently participate in off-farm or part-time activities such as small-scale commerce or construction to augment income during periods of agricultural instability [27].

Sustainable income diversification is essential for long-term food security, especially in areas hit by climate change and economic instability. By mixing climate-smart farming with non-farming income, households can reduce their reliance on resource-draining practices and become more resilient to unexpected challenges [28]. This approach not only helps balance the environment but also provides a steady income, which is crucial in places dealing with environmental pressures like droughts and floods [6,29].

Food security is a complex concept characterized by reliable physical, social, and economic access to sufficient, secure, and nutritious food that fulfills individuals’ dietary needs for an active and healthy lifestyle. This notion has four fundamental dimensions: availability, utility, access, and stability [1]. These factors are essential for recognizing how various revenue sources might affect households’ abilities to obtain food.

The influence of income diversification on food security is broadly recognized across many socio-economic contexts. Research in Ghana’s Upper East and Upper West regions reveals that engagement in non-farm employment markedly enhanced household financial stability and food security [30]. Similarly, diversifying income, facilitated by NGOs and the Ghana Ministry of Food and Agriculture, is crucial for alleviating poverty and improving food security. Furthermore, rich families with varied income streams had significant enhancements in both income and caloric consumption. The study found that female-headed families and those with focused assets had further difficulties, since personal risk perceptions affected diversification choices [28].

Households involved in aquaculture, livestock husbandry, or non-agricultural employment generally sustain better reserves of food during economic recessions or environmental disturbances, especially in areas characterized by poor agricultural output [31]. Furthermore, non-agricultural initiatives, including wage labor and small enterprises, play a vital role in ensuring food availability in regions where agricultural production is insufficient to satisfy food requirements [32]. Income diversification improves food availability by supplying households with consistent income sources, allowing them to participate in many food markets and acquire a broader selection of healthy foods [33]. The augmented purchasing power derived from diversified income enhances food use, enabling households to purchase diverse and high-quality meals crucial for optimal health and nutrition [29,34]. Diversified income sources enhance food stability by offering financial resilience against shocks such crop failures, price volatility, or income interruptions, thus assuring a stable food supply year-round [35]. Access to key agricultural inputs like quality seeds, fertilizers, and irrigation is essential for boosting productivity. However, in places like Afghanistan, where infrastructure is weak and markets are hard to reach, farmers struggle to obtain these vital resources. This not only limits food availability but also makes households more vulnerable to external shocks. To cope, many Afghan families have turned to income diversification. By engaging in non-agricultural activities, they gain the financial stability needed to deal with poor harvests and shortages of essential inputs [24,36].

Government policies are crucial in facilitating diversification of income by improving access to finance, advancing infrastructure, and reinforcing market connections. Enhanced access to government programs and financial resources might motivate people to engage in various income-generating activities, thereby improving food security [28]. Effective policies must prioritize training and skill development to facilitate family diversification into sectors other than agriculture, resulting in more predictable revenue streams [1]. Furthermore, implementing equitable market pricing and facilitating market access might enhance the sustainability of income diversification initiatives, therefore mitigating sensitivity to food insecurity [37].

In conflict-affected areas, food geopolitics and trade policies often make it harder to access essential agricultural resources and markets. Global trade barriers, political instability, and international policies create obstacles to food availability, especially for rural families who rely on imports. In Afghanistan, these geopolitical issues have made the food supply even more unstable, causing prices to fluctuate and limiting access to important farming inputs. As a result, households in regions like Helmand are increasingly turning to off-farm income, migration, and remittances to cope with growing food insecurity [6,36].

Empirical research from several locations elucidates how income diversification improves food security. In Ethiopia, households participating in non-agricultural occupations during periods of agricultural shortage are more capable of acquiring food and sustaining nutritional stability, hence diminishing their reliance on agricultural revenue during lean seasons [38]. Research in Southern Ethiopia demonstrates that crop diversification, especially into vegetables and high-value crops, substantially increases family revenue and variety in diet, thereby enhancing the availability of food during periods of agricultural stress [39]. In Pakistan, diversifying income through other than agriculture employment, including emigration, has proved essential for enhancing household resilience to climate-related threats. These solutions enhance household income and serve as a safeguard amid food shortages resulting from environmental difficulties [40]. In Bangladesh, paid work and wage labor enhance food security, allowing households to manage seasonal agriculture variations more efficiently [41]. Smallholder farmers in Latin American countries such as Colombia and Mexico, who practice diversifying their crops and engage in non-farm activities, exhibit enhanced resilience to climate variability and economic shocks, thereby improving income and food security [42].

Gender dynamics profoundly affect income diversification, since households with women in charge frequently face obstacles in obtaining non-farm sources of income. Women, especially in rural agricultural households, significantly contributed to off the farm income [43]. Nonetheless, inequities in education, resource accessibility, and career prospects frequently impede women’s capacity to diversify successfully [44]. Implementing targeted interventions to address gender-related barriers might enhance food security results by enabling women to engage more actively in various income-generating activities. Technological improvements, including mobile banking, digital platforms, and agricultural developments, have improved chances for income diversification, particularly in rural regions [45]. Mobile money services have facilitated households in engaging in off-farm activities, accessing remittances, and participating in trade more effectively, hence enhancing food security [46]. Incorporating technology into diversification plans is essential for improving the availability of food and mitigating vulnerabilities to shocks.

Although diversifying one’s income provides significant advantages, it also entails obstacles and constraints. Wealthy households may be less motivated to diversify their revenue streams due to the substantial operating expenses linked to non-farm activities [47]. Moreover, government subsidies alone may be inadequate for enhancing food security without the implementation of financial security mechanisms, such as basic income guarantees [48]. Consequently, efforts aimed at improving diversification of income must be complemented with supporting policies to optimize their effect on food security.

Climate change presents a substantial challenge to conventional agricultural practices, rendering diversifying income an essential risk management technique [5]. As climatic unpredictability escalates, families are increasingly engaging in non-farm activities to protect their livelihoods and food security [49]. Advocating for climate-resilient farming methods and alternative revenue streams is crucial for improving family adaptive capacity and guaranteeing sustainable food security in the face of environmental problems. Climate unpredictability in regions like Afghanistan highlights the need for income diversification as a risk management strategy. With agricultural yields affected by climate-induced droughts and floods, households in areas like Helmand increasingly depend on non-agricultural income to maintain food security. Research shows that such diversification helps families cope with climate-related agricultural disruptions, ensuring nutritional stability [19].

Socio-demographic characteristics, including the age of the family head, educational achievement, and family size, substantially affect food security results. Research in Mexico and Bangladesh has shown that these characteristics are essential in influencing food security, with educated families exhibiting greater resilience [50,51]. The value of product sold, education, access to financing, and involvement in public assistance programs as key drivers of food security among maize farmer families in Nigeria [52].

Despite the extensive research on income diversification and food security, gaps remain, particularly in conflict-affected and environmentally vulnerable countries such as Afghanistan. This study aims to address these gaps by examining the association between income diversification and household food insecurity in Helmand Province, focusing on how households utilize income diversification strategies to enhance food security amid economic instability, environmental degradation, and ongoing conflict. The study explores socio-economic, bio-socio-cultural, and locational factors influencing income diversification and assesses their effectiveness in mitigating food insecurity. By doing so, the research will contribute to a deeper understanding of sustainable livelihoods in fragile, conflict-affected, and food-insecure regions, offering valuable policy recommendations to support household resilience and long-term food security, particularly in developing countries.

### 1.2. Theoretical Concepts

#### 1.2.1. Income Diversification and Food Insecurity

Income diversification involves securing income from multiple sources, both within and outside of agriculture, to reduce household reliance on a single income stream [28]. This approach is crucial for building resilience against food insecurity, especially in conflict-affected areas where households face economic instability, environmental shocks, and limited food access [53]. Diversifying income-generating activities across sectors helps households withstand unpredictable threats, improving their ability to cope with disruptions and ensuring more stable food access [54].

In Afghanistan, especially in rural regions, households often depend on a mix of agricultural activities, such as crop cultivation, livestock farming, and integrated farming systems, alongside non-farm income sources, including wage labor, small businesses, handicrafts, and remittances [8,22]. Income diversification strengthens households’ abilities to weather environmental (e.g., droughts) and economic (e.g., market fluctuations) shocks, ensuring a more stable food supply despite unexpected challenges [55].

Income diversification boosts households’ abilities to cope with shocks by providing multiple income sources. Households engaged in non-agricultural activities, like trade or construction, typically experience more stable and higher returns, especially in areas where agriculture is limited by environmental or market factors [26]. In Afghanistan, where agricultural risks are exacerbated by climate change and conflict, non-farm income sources play a critical role in bolstering food security, helping to mitigate the impacts of these compounded challenges [31].

Income diversification boosts household income and strengthens food security by ensuring availability, access, utilization, and stability [34]. By diversifying their income sources, households improve their purchasing power, enabling them to afford a broader range of nutritious foods [56]. Furthermore, diversified income plays a crucial role in stabilizing food access during periods of scarcity, protecting families from the fluctuations and uncertainties often accompanying agricultural production cycles [57].

#### 1.2.2. Conceptual Framework

The conceptual framework of this study is based on the idea that income diversification plays a key role in shaping food insecurity, with moderating factors, such as socio-economic, bio-socio-cultural, and locational factors, affecting how strong or weak this relationship might be. The framework looks at how these moderating factors interact and either strengthen or limit the effect that income diversification has on food security in Afghanistan.

Income diversification refers to earning money from different sources rather than depending on just one. This is especially important for households that face economic challenges, as having multiple income streams provides a safety net [8]. Food insecurity, however, means not having reliable access to enough affordable, nutritious food to live a healthy life [2].

Socio-economic factors, like household income, education, and family size, are crucial in determining whether a household can diversify its income sources. Households with better education and more financial resources are typically more able to participate in various income-generating activities, which helps them become more resilient and secure in terms of food access [43,44].

Bio-socio-cultural factors, such as gender roles and cultural attitudes, also influence how much a household can diversify its income. In Afghanistan, women face significant challenges in accessing non-agricultural income sources due to societal restrictions and limited access to education and resources [58]. These gender-based barriers must be considered when looking at the role of income diversification in improving food security, as they can reduce the effectiveness of diversification strategies in specific households.

Locational factors, such as infrastructure, market access, and geographic location, are also important. Households that live in areas with better infrastructure and closer access to markets are more likely to succeed in diversifying their income sources and, in turn, improving their overall food security [59].

These factors determine how much income diversification can help reduce food insecurity. Figure 1 shows how these factors connect, illustrating how income diversification, socio-economic factors, bio-socio-cultural factors, and locational characteristics influence food security in the study area. This framework assumes that these moderating factors influence the relationship between income diversification and food insecurity.

## 2. Materials and Methods

The methodological diagram (Figure 2) outlines the study’s multistage random sampling approach across five districts in Helmand Province, Afghanistan, which included 302 households. Data analysis employed descriptive statistics, ANOVA, chi-square tests, and ordered logistic regression to assess how socio-economic, bio-socio-cultural, and locational factors influence household food insecurity and income diversification.

### 2.1. Study Area

Helmand Province, located in southern Afghanistan, spans 58,584 square kilometers and has 13 districts with over 1000 villages [36]. This study focuses on the districts of Nad Ali, Nawa-e-barakzaiy, Lashkargah, Nahar-e-Saraj, and Garmser, chosen for their proximity to the provincial capital, Lashkargah, and their relevance to agricultural activities (Figure 3).

Agriculture is the primary income source for households in Helmand, followed by other livelihood activities such as manual labor, services, manufacturing, and trade [60]. The 1980s irrigation infrastructure supported double cropping, such as maize, barley, mung beans, and wheat [61]. Livestock farming is also integral to household economies, with families raising cattle, sheep, goats, donkeys, and chickens for milk, meat, eggs, and transportation; surplus poultry products, particularly eggs, are often sold by women [36]. Due to the region’s favorable climatic conditions, double cropping, cultivating both winter and summer crops on the same land, is widely practiced [61].

However, Helmand’s agricultural sector faces numerous seasonal challenges. Climate change has reduced water availability, affecting productivity and pushing households to diversify their income sources [19]. These environmental pressures are compounded by socio-economic limitations, including seasonal unemployment, labor market constraints, and large household sizes averaging nearly 13 members, which collectively hinder agricultural productivity [22,24]. Socio-demographic barriers further exacerbate these challenges.

The province has a population of 1,525,188, with 93% living in rural areas, mostly Pashtun tribes [62]. Low literacy, especially among women, limits employment and income diversification. Households rely on off-farm work, livestock sales, and remittances to cope with food insecurity and economic instability [19,22].

Despite these challenges, households in various regions of Afghanistan have adopted income diversification strategies, such as small-scale commerce and handicrafts, to reduce agricultural risks and improve food security [22]. However, these efforts are hindered by inadequate infrastructure, limited financial access, and conflict [19,63]. Given these vulnerabilities, Helmand represents a critical context for studying how income diversification interacts with food insecurity and the sustainability of rural livelihoods.

### 2.2. Sample Size and Sampling Approach

A cross-sectional study was conducted in five districts of Helmand Province using a well-designed questionnaire and multistage random sampling—the sample comprised household heads. Primary data were collected on income diversification, food insecurity, and factors related to (a) socio-economic (occupation, education, income, household size, land ownership, and financial status), (b) bio-socio-cultural (age, gender, marital status, family composition), and (c) locational factors (distance to district bazaar and capital city, transport access). Sample size was determined using Cochran’s formula based on confidence level, margin of error, and population proportion [64].(1)$$n=\frac{Z^{2}*p.*\left(1-p\right)*N}{E^{2}*\left(N-1\right)+Z^{2}*p*\left(1-p\right)}=302$$ where N is the total population of Helmand, *p* = 0.5 is the proportion for estimate, E = 0.05 is the margin of error, Z is the Z-score for 95% confidence, and n is the sample size of 302.

### 2.3. Procedure for Data Collection

The study was carried out in 2024, concentrating on Helmand Province, which has a population of 1,525,188. The rural populations of Lashkargah, Nad Ali, Nawa-e-barakzaiy, Nahar-e-Saraj, and Garmser were documented as 110,778, 196,866, 117,174, 172,450, and 32,042, respectively, while the urban populations in Lashkargah and Nahr-e-Saraj districts were 95,883 and 11,893, respectively [62]. A total of 302 rural households were randomly selected from five purposively chosen districts in Helmand Province (Nawa, Garmsir, Nahr-e-Saraj, Lashkargah, and Nad Ali), selected due to their proximity to the provincial capital, better accessibility, logistics, and higher population coverage (over 48%), which influence socio-economic factors relevant to the study. Each district contains four to six administrative units (towns), from which two units were randomly selected to balance representativeness and logistical feasibility. Within each selected unit, three villages were randomly chosen to capture the area’s diversity. From each village, 10 households were randomly selected, except in two villages where 11 were chosen to reach the total of 302, while ensuring equal selection probability to reduce bias. This sample size and multi-stage random sampling were designed to ensure statistical power and represent the area’s socio-economic variation. All field investigators underwent comprehensive training before collecting data to ensure data quality and minimize potential errors. This training covered interviewing techniques, ethical considerations, and standardized procedures for recording responses accurately. Additionally, pre-testing of the survey instrument was conducted to identify and rectify any inconsistencies.

### 2.4. Measurement of Variables

#### 2.4.1. Dependent Variable: Household Food Insecurity

The food security variable is derived from the combination of responses to nine questions and their frequency, resulting in a four-tier food insecurity scale: food-secure (defined “1”), mildly food-insecure (defined “2”), moderately food-insecure (defined “3”), and severely food-insecure (defined “4”). The categories are defined by the following cut-off conditions:HFIA category = 1 if [(Q1a = 0 or Q1a = 1) and Q2 = 0 and Q3 = 0 and Q4 = 0 and Q5 = 0 and Q6 = 0 and Q7 = 0 and Q8 = 0 and Q9 = 0]HFIA category = 2 if [(Q1a = 2 or Q1a = 3 or Q2a = 1 or Q2a = 2 or Q2a = 3 or Q3a = 1 or Q4a = 1) and Q5 = 0 and Q6 = 0 and Q7 = 0 and Q8 = 0 and Q9 = 0]HFIA category = 3 if [(Q3a = 2 or Q3a = 3 or Q4a = 2 or Q4a = 3 or Q5a = 1 or Q5a = 2 or Q6a = 1 or Q6a = 2) and Q7 = 0 and Q8 = 0 and Q9 = 0]HFIA category = 4 if [Q5a = 3 or Q6a = 3 or Q7a = 1 or Q7a = 2 or Q7a = 3 or Q8a = 1 or Q8a = 2 or Q8a = 3 or Q9a = 1 or Q9a = 2 or Q9a = 3]

We derived our dependent variable, food insecurity, using the nine-item binary outcome (“yes = 1” or “no = 0”) Household Food Insecurity Access Scale (HFIAS) [65].

The nine questions gather responses on concerns about household food security, perceptions of food availability and quality, reductions in food intake, and the consequences of food insecurity. Each of the nine questions includes a frequency scale: seldom, sometimes, and often (see “Appendix A” for details) [65]. While defining food insecurity as a categorical variable could cause worries regarding thresholds, it facilitated the comparison of relationships, as delineated in other research e.g., [66,67]. Food insecurity is defined as insufficient access to inexpensive, nutritious food, resulting in issues related to hunger and malnutrition [68].

#### 2.4.2. Independent Variables: Household Income Diversification and Control Variables

Previous empirical research, including [3,69,70,71], has emphasized the significance of deficiency, climatic conditions, and socio-cultural variables in influencing access to livelihoods and food. This research defines income diversification as the number of income sources families report. This metric is based on replies to 23 questions concerning yearly household income, with answers encoded as “yes” (1) or “no” (0). The study encompasses a range of income-generating activities and assets, including sales of agricultural and non-agricultural products, pensions, business ventures, informal labor, government and NGO initiatives, revenue from permanent and temporary work, remittances, and monetary gifts. A “yes” response means that the household obtained money from that source in the year prior to the survey. Based on all twenty-three responses, an additional score assessed the diversity of household income sources. A score of “0” indicates negative responses to all questions, while a score of “23” reflects positive responses to every question. Lower scores indicate less income diversification, while higher scores suggest greater diversification. This approach to quantifying income diversity, based on the number of income sources, aligns with methods used in other studies e.g., [72,73]. Income diversification refers to the expansion of income sources for individuals or households [28] and is recognized for its ability to reduce risks and strengthen resilience against agricultural challenges [8,74].

Using the sustainable livelihood framework [9] and the entitlement theory [75], while drawing from the literature on income diversification and food security, we identified relevant covariates for analysis, classifying them into socio-economic, bio-socio-cultural, and locational factors [76]. Socio-economic variables affect unequal access to food, while bio-socio-cultural factors are influenced by socio-cultural contexts that either support or restrict food production and access [75]. Similarly, locational factors affect food production and distribution capabilities, significantly influencing availability and access disparities.

### 2.5. Analytical Approaches

The study used descriptive statistics and inferential tests, including chi-square tests and ANOVA, to evaluate the statistical significance of differences across categories. Additionally, logistic regression analysis was undertaken to investigate the relationship between income diversification and household food insecurity status. Ordered logistic regression was used because of the categorical and ordinal nature of the outcome variable (food insecurity), which ranges from food-secure to severely insecure. The specifications for the ordinal logistic regression model are as follows:(2)$$\log[\frac{P(Y_{ij}\leq 1)}{\left(1-P(Y_{ij}\leq 1\right)}]=\alpha_{0}+\sum_{k=1}^{p-1}\alpha_{jk}X_{ijk}+V_{ij},C=1,\ldots \ldots .\Omega -1$$ where *P*(*Y*_ij_ ≤ 1) denotes the probability that the given event will occur, [(1 − *P* (*Y_ij_* ≤ 1)] represents the probability that the event will not occur, *α_jk_* is the coefficient terms, *X_ijk_* are the explanatory variables, *k* = 1 is the first explanatory variable, and *p* − 1 is the last explanatory variable. *α*_0_ and *Ω* − 1 are the intercept terms, and *V*_ij_ represents the error term in the logistic model [77].

Households are more likely to prefer being in the food-secure category, which represents a lower level of food insecurity compared to the moderately or severely food-insecure categories [65]. The maximum likelihood estimation method was employed to calculate the odds ratios for reporting high levels of food insecurity [78]. An odds ratio below 1 indicates that families exhibiting that characteristic are less likely to experience severe food insecurity than those categorized as moderate or secure. An odds ratio above one indicates that families showing this characteristic are more vulnerable to experiencing severe food insecurity [79].

Before conducting the regression analysis, a multicollinearity assessment was performed using SPSS version 25, which calculated the Variance Inflation Factor (VIF) values for each variable. The average VIF was found to be 2.1, indicating a moderate level of collinearity among the predictors. Since VIF values below 5 typically do not suggest problematic multicollinearity, this suggests that multicollinearity is unlikely to significantly affect the results of the regression. Additionally, logistic regression assumes that observations are independent of each other.

Three stages of analysis were conducted to answer the study issue thoroughly. Firstly, descriptive statistics were used to draw out the sample. Secondly, bivariate ordered logistic regression was performed to investigate zero-order correlations between independent factors and household food insecurity. Three multivariate nested models were constructed to investigate the independent correlation between income diversification and household food insecurity and examine the impact of socio-economic, bio-socio-cultural, and locational variables on this association. All analyses were conducted utilizing SPSS software, version 25.

## 3. Results

### 3.1. Descriptive Statistics of Household Food Insecurity and Income Diversification in Helmand, Afghanistan

Table 1 provides a study of households’ food insecurity in Helmand, Afghanistan, based on a sample of 302 households. The data are classified into four categories of food insecurity: food-secure (5.3%), mildly food-insecure access (8.6%), moderately food-insecure access (22.5%), and severely food-insecure access (63.6%).

Households predominantly headed by farmers (65.3%, *p* < 0.001), particularly those impacted by illiteracy (91.1%, *p* < 0.001), low yearly income (66.8%, *p* < 0.001), low-income diversification (mean 1.6, *p* < 0.001), large family sizes of more than 10 members (83.2%, *p* < 0.000), a reduction in income compared to the previous year (94.9%, *p* < 0.001), absence of land ownership (95.5%, *p* < 0.000), and those with struggling financial situation (82.6%, *p* < 0.001), demonstrate serious vulnerability to severe food insecurity. These households generally inhabit temporary shelters (93.1%, *p* < 0.001), lack relatives employed by the government (83.9%, *p* < 0.000), and have limited access to transportation (69.7%, *p* < 0.000), which worsens their vulnerability. The prevalence of food insecurity in these families is notably higher among married couples, with (79.4% *p* < 0.001) of such households categorized as severely food-insecure. Furthermore, households with a greater percentage of female members (mean = 9.1, *p* < 0.001), a higher count of children under 16 years (mean = 8.6, *p* < 0.000), a small farm size (mean = 0.7, *p* < 0.000), and those situated at a greater distance from district bazaars (mean = 24.5, *p* < 0.016) and capital towns (mean = 76.8, *p* < 0.001) are more affected.

Conversely, families that diversify income sources, especially those with jobs outside of farming and a higher overall income, tend to be more food-secure. Smaller family sizes, an increase in income compared to the previous year, having more male members in the household, and better access to transportation also play a key role in reducing food insecurity. Together, these factors help make households more financially stable and better able to meet their food needs.

### 3.2. Household Food Insecurity Access Scale (HFIAS)

Table 2 indicates severe food insecurity among the respondents. A significant 91.72% of respondents expressed concern about food shortages, with 38.40% reporting this rarely, 29.80% sometimes, and 23.50% often. Furthermore, 88.74% of respondents could not eat whenever they wanted, with 39.10% rarely, 25.80% sometimes, and 23.80% often. There was also a reported decline in meal variety and quality for 88.10% of households, with 87.10% resorting to consuming undesirable meals, 83.40% experiencing insufficient lunches, and 75.50% limiting food quantities.

Additionally, 62.30% of households reported that they sometimes have no food, with 21.20% indicating this occurred rarely, 26.20% sometimes, and 14.90% often. Overall, 54.60% went to bed hungry (23.20% rarely, 19.20% sometimes, 12.30% often), and 58.90% experienced an entire day and night without eating (28.10% rarely, 20.50% sometimes, 10.30% often). The frequency breakdown highlights that these challenges are recurring, underscoring chronic food insecurity in the region.

### 3.3. Income Sources

Table 3 shows the different income sources that households have relied on over the past 12 months, highlighting the primary means of financial support for households. The most common source of income is contract farming, with over a quarter (28.3%) of households reporting income from this activity, emphasizing its important role in the local economy. Other major income sources include animal products (19.4%), staple grains (9.7%), and vegetables (6.6%), which underscore the importance of agriculture in supporting household income. Additionally, self-employment (6.6%) and temporary employment (6.1%) play an important role, reflecting the dominance of informal employment in the region. Handicraft production and garden fruits also contribute to household income, reported by 4.2% and 3.7% of households, respectively. The study reveals that agriculture, livestock, and small-scale informal employment are the primary sources of income for many households.

Smaller income sources include poultry products (2.4%), seasonal migration (1.8%), and remittances (1.6%), showing economic diversification among households. Additional income comes from permanent employment, fishery products, and NGO programs, each providing income to around 1.5% of households. Less common sources of income include small-scale trade and government programs (each 0.8%), as well as lending farmland and renting shops or houses (each 0.6%). Income from vehicles for transport, tractors, and threshers is reported by less than 1% of households. Notably, there were no reports of pension income, which suggests limited access to formal social welfare. Overall, the finding underscores the heavy reliance on agriculture, informal jobs, and family-based income sources, with formal financial assistance playing a relatively small role in supporting households.

### 3.4. Bivariate and Multivariate Ordered Logistic Models Predicting Severe Food Insecurity

The findings for both bivariate and multivariate analyses are displayed in Table 4. At the bivariate level, families exhibiting more income diversity (number of income sources) were associated with a reduced likelihood of food insecurity, suggesting that income diversification contributes to improved food security. Specifically, families with diverse income streams were more food-secure compared to those experiencing mild, moderate, and severe food insecurity (OR = 0.62, *p* ≤ 0.001). This suggests that diversified income sources provide financial stability, enabling households to buffer against economic shocks and disruptions in agricultural production.

Most socio-economic characteristics were associated with an increased likelihood of food insecurity. For example, households led by farmers had a greater likelihood of severe food insecurity than other occupation types (OR = 4.073, *p* ≤ 0.05). Similarly, households in lower- and middle-annual-income groups, compared to those in higher-income categories and those with less and the same income as the previous year, exhibited increased odds of reporting severe food insecurity. Compared to homeowners, this trend was observed among individuals residing in rented accommodations, living with family members, or in temporary shelters. Additionally, those without land ownership, lacking relatives in government employment, and experiencing bad financial situations were similarly more likely to report severe food insecurity. These results underscore the significant role that socio-economic factors play in influencing food security. We also discovered that locational and bio-socio-cultural characteristics were significant determinants of household food insecurity.

Upon adjusting for socio-economic characteristics in model 1, the association between income diversification and lower odds of household food insecurity remained strong, with a little reduction in the likelihood of reporting severe food insecurity (OR = 0.50, *p* ≤ 0.001). Households with the same income as the previous year were five times more food-insecure than those with more income (OR = 5.46, *p* ≤ 0.001). The significance of variables such as main occupation, annual income, and resident tenure status disappeared, indicating that income diversification may mitigate their impact. However, socio-economic factors such as financial situation, lack of land, and households headed by people who are illiterate remained critical predictors of severe food insecurity.

In model 2, the association between income diversification and reduced chances of severe food insecurity remained significant after adjusting for bio-socio-cultural characteristics, with a slight drop in the risks of severe food insecurity (OR = 0.52, *p* ≤ 0.001), indicating that bio-socio-cultural factors, such as household gender composition and family size, influence food insecurity dynamics. Furthermore, financial status remained a significant factor, consistent with model 1. Smaller households with more male family members and larger farm sizes displayed reduced food insecurity. Conversely, having no land, a household headed by someone who was illiterate, not having a relative employed in government, and having the same income as the previous year significantly increased the likelihood of severe food insecurity. This suggests that bio-socio-cultural factors are critical predictors of household food insecurity.

Lastly, after introducing locational factors in model 3, the association of income diversification and lower odds of food insecurity remained significant (OR = 0.49, *p* ≤ 0.001). Household characteristics such as size, farm size, and gender composition (more male members) consistently lowered food insecurity risk. However, socio-economic and bio-socio-cultural factors such as households headed by someone who was illiterate, those with the same income compared to the previous year, those having no land, those with an absence of relatives in government jobs, those with a struggling financial situation, more female family members and distance from distract bazaars were more likely to report being severely food-insecure. These findings underscore socio-economic, bio-socio-cultural, and locational characteristics that significantly shape household food insecurity dynamics, even in income diversification.

Together, these findings suggest that socio-economic, bio-socio-cultural, and locational factors significantly influence household food insecurity, and income diversification remains an important strategy for mitigating food insecurity.

## 4. Discussion

This study reveals a strong association between income diversification (i.e., the number of income sources) and reduced likelihood of severe food insecurity in Helmand, Afghanistan. The study observed that households with more income sources are less likely to face severe food insecurity, suggesting that diversification serves as a crucial strategy for enhancing food security in Afghanistan, as supported by [63], who argue that households with different income streams exhibit greater resilience to economic fluctuations and environmental disruptions, thereby reducing their vulnerability to food shortages. Similarly, households with diversified income sources are more resilient to shocks such as market fluctuations and environmental disruptions, underscoring income diversification’s protective role in enhancing food security [80].

Socio-economic adjustments in this study slightly mitigate the effect of income diversification on food security, suggesting that socio-economic factors partially moderate this relationship. The result demonstrates that households with better socio-economic conditions are better equipped to leverage income diversification strategies to reduce food insecurity. For example, educated households with stable income sources are better able to manage food access, even in the face of external shocks. This finding is consistent with research conducted by [9]. Furthermore, Lisa et al. [81] found that socio-economic factors, such as increased income and land access, enhance households’ abilities to diversify income and meet nutritional needs. Diversified livelihoods help rural households manage risks, thereby boosting food security and coping capacity in challenging economic conditions [82].

Further analysis combining bio-socio-cultural factors emphasizes their significant moderating impact on food security. This shows the importance of family structure, gender dynamics, and cultural practices in shaping food security outcomes. For example, male-headed households benefit from additional agricultural labor, while female-headed households face more barriers to accessing resources, exacerbating food insecurity. This finding is align with research conducted by [83].

Locational factors further modify the relationship between income diversification and food insecurity, indicating that households located in remote areas are less able to diversify their incomes, which increases their susceptibility to food shortages. Households near to markets play a crucial role in mitigating food insecurity, emphasizing the need for infrastructure development and improved market access in rural areas to support income diversification and food security. This is in line with the research conducted by [84].

Household income diversification and food security are also influenced by various socio-economic, bio-socio-cultural and locational factors, including education, household size, income compared to the previous year, land ownership, relative holds government Job, financial situation, farm size, male family members, and distance from district bazaars.

Education plays a crucial role in reducing food insecurity, as illiterate households are significantly more likely to experience severe food insecurity than educated households. This suggests that education is essential for building resilience and addressing food insecurity, particularly in vulnerable communities. This study’s findings support the conclusions drawn by [85,86], which emphasize education as an important factor in improving food security outcomes. Education not only increases income potential and job opportunities [1] but also empowers individuals to make better-informed decisions about agriculture, diet, and managing resources [81].

Smaller households (10 or fewer members) are significantly less vulnerable to severe food insecurity than larger households (more than 10 members). This reveals that smaller households are able to manage food resources more efficiently due to less competition for those resources. These results align with [87], who notes that larger families in low-income settings face greater food insecurity due to increased strain on resources. furthermore, Larger households may adopt coping strategies like reducing meal frequency and worsening food insecurity [6].

Households whose income remained the same compared to the previous year are more likely to experience food insecurity. The finding underscores that income stagnation contributes to food insecurity, primarily due to rising costs, inflation, and unexpected expenses. This is consistent with previous studies on the relationship between income stability and food security. Even when income remains constant, it may be insufficient to meet rising basic needs, thereby worsening food insecurity [88]. Similarly, Coleman-Jensen. [89] found that stagnant income predicts food insecurity in the U.S. as households struggle with inflation. Likewise, stable or declining incomes, especially during downturns, worsen European food insecurity [90]. In line with these findings, Nord. [91] confirms that the absence of income growth over time is a key factor in the increased likelihood of households becoming food-insecure.

Land ownership is another crucial determinant of food security, as households without land ownership face significantly greater risks of food insecurity. This highlights the importance of land access programs and tenure security in mitigating food insecurity, particularly in rural and agrarian settings. In this context, land ownership provides resources and stability, allowing food production and income generation, thus reducing vulnerability [92]. In contrast, landless households, dependent on unstable income sources, are more food-insecure [93]. Similarly, landless households face challenges in accessing loans, inputs, and markets, which worsens their vulnerability to food insecurity [94].

A relative employed in a government post is a substantial safeguard against acute food insecurity. The finding indicates that government employment plays an important role in reducing the likelihood of severe food insecurity, particularly in low-income households. This underscores the importance of expanding access to stable public sector employment as a strategy to enhance household resilience against food insecurity. The present result supports the work, which found that government jobs offer more consistent income, benefits, and job security than informal or agricultural work, particularly in low-income areas. In addition to providing financial stability, public sector employment also provides socio-economic stability, shielding households from economic shocks that can lead to food scarcity [95]. Furthermore, rural households often face greater food insecurity due to limited stable work and market access, which secure government jobs can mitigate by ensuring reliable income and social services [96].

Financial stability plays a crucial role in the association with food security. The finding implies that financial stability significantly alleviates food insecurity, with income stability particularly influencing low-income households. Households with stable income are better able to manage trade-offs between food and other essential needs, as a strong association exists between families’ perceptions of financial stability and food security [88]. Furthermore, financial resources help households access nutritious food, manage price fluctuations, and invest in long-term food security [81]. Additionally, income levels are identified as a key factor, emphasizing the need for policies that improve income support and reduce poverty to mitigate food insecurity [89].

Farm size is consistently associated with reduced food insecurity. The findings suggest that farm size plays a critical role in mitigating food insecurity, with larger farms offering greater protection against economic and environmental stressors. These results emphasize the importance of supporting smallholder farmers and enhancing access to land and resources as key strategies for improving food security. This finding aligns with previous research, which has shown that access to larger agricultural holdings allows households to meet food needs and generate income, safeguarding against food insecurity [97,98]. Larger farms also enhance resilience to external shocks, such as weather and market fluctuations, thereby reducing vulnerability to food insecurity [99].

The presence of male family members is associated with reduced severe food insecurity, highlighting the role of male labor in income and food security in patriarchal contexts. Each additional male family member reduces the likelihood of severe food insecurity, as they contribute labor and income in resource-limited settings [100].

In contrast, the presence of additional female family members in a household is associated with a higher risk of severe food insecurity due to gendered resource distribution and economic vulnerability. When women lack control over finances and decision-making, resource strain increases, worsening food insecurity, especially in households with multiple women [101]. In Arab countries, gender differences in income and employment worsen this, as women often prioritize family needs during economic crises [102]. Potential solutions involve improving women’s access to income, promoting gender equity, and strengthening social safety nets [102].

Finally, proximity to district bazaars is a significant determinant of food security. The finding demonstrates that households located farther away from district bazaars face challenges accessing food, which could be due to higher transportation costs, limited market options, or reduced trade opportunities. The study is consistent with the work, who states that easier market access improves buying and selling, reducing food poverty risks [84]. Similarly, geographical isolation restricts market access, worsening food poverty, especially in rural areas [96]. On the other hand, proximity to markets access enhances trade opportunities and provides access to diverse range of food sources, thereby reducing vulnerability [103].

## 5. Conclusions

The study highlights the crucial role of income diversification in reducing food insecurity in Helmand, Afghanistan. Using both bivariate and multivariate models, the findings reveal that households with lower levels of income diversification are much more vulnerable to severe food insecurity, highlighting the need for income diversification as a key strategy. The research also highlights how socio-economic, bio-socio-cultural, and locational factors influence food insecurity, demonstrating its complex nature. Households facing economic challenges, such as not owning land, financial difficulties, illiterate heads of households, or no relatives in government, were more likely to experience food insecurity. On the other hand, smaller household sizes, a higher proportion of male members, and larger farm sizes were linked to better food security. These findings suggest that while income diversification is important, it is not a standalone solution. Its effectiveness is shaped by various socio-economic and contextual factors, all of which need to be addressed for meaningful improvements in food security.

However, it is essential to acknowledge limitations, such as the reliance on cross-sectional data, which restricts causal inferences, and focus solely on Helmand, which may not fully reflect the broader context of food insecurity across Afghanistan. Future studies should explore longitudinal impacts and apply similar modelling techniques across diverse regions in Afghanistan to provide a broader understanding of food insecurity dynamics. Research exploring the temporal variations in income sources and food insecurity and qualitative data to understand better the underlying mechanisms would offer deeper insights into the causal processes and moderating factors affecting food security. It is also important to note that the Household Food Insecurity Access Scale uses a four-week recall period, while income diversification is based on a twelve-month recall. This difference in recall periods could introduce bias into the estimated results.

To address food insecurity effectively, targeted interventions should address economic challenges by promoting income diversification and supporting women in income-generating activities, such as livestock raising (e.g., goats, sheep, chickens, and ducks), home gardening (e.g., vegetables, fruits, and herbs), and handicrafts (e.g., Kandahari hats, embroidery). Training programs in tailoring and small-scale food processing (e.g., drying fruits or producing jams) could further enhance women’s livelihoods, providing nutritional and economic benefits.

The government should promote land reform policies, ensuring equitable access to land for landless farmers and marginalized groups. Additionally, collaborating with banks and microfinance institutions to offer loans or profit-sharing financing based on Islamic finance principles can help rural households diversify their income and increase productivity. Access to modern farming techniques, like water-efficient irrigation, drought-resistant seeds, better soil management, and support for sustainable fish farming and beekeeping, would significantly boost productivity. Extension services are essential to support farmers, fish farmers and beekeepers in adopting best practices and new technologies for long-term success. Microfinance for small businesses and equipment could increase income and reduce post-harvest losses, while low-cost storage technologies, like grain silos, would minimize food waste.

Short workshops for men on family planning and managing resources can help families better manage their household size and resources, improving food security. Education, particularly adult literacy and skills training programs for illiterate household heads, would reduce vulnerability. Expanding market access for agricultural products, strengthening supply chains, and enhancing local market capacity could increase income. Vocational training in agriculture and entrepreneurship and promoting cooperatives would further boost farmers’ bargaining power and market access. A multi-faceted approach addressing socio-economic, bio-socio-cultural, and locational factors is necessary to enhance household resilience and improve food security in Helmand and similarly conflict-affected and economically challenged regions. Such an approach will ensure that interventions are tailored to the unique needs of different communities, ultimately contributing to more effective and sustainable solutions for food insecurity.

## Figures and Tables

**Figure 1 foods-14-02048-f001:**
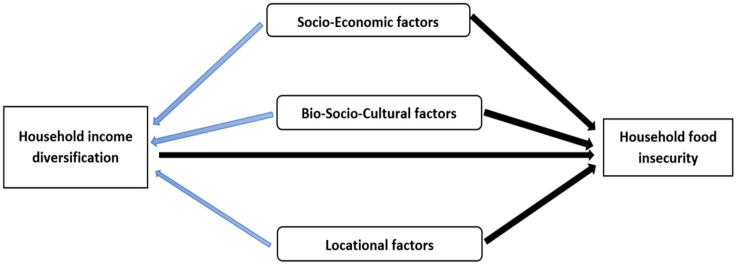
Research conceptual framework.

**Figure 2 foods-14-02048-f002:**
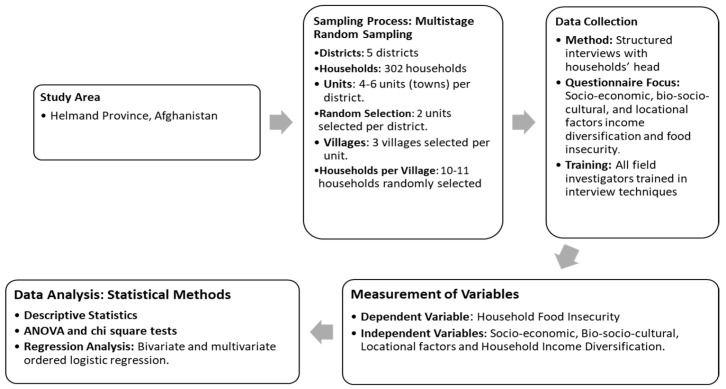
Methodological diagram of the study.

**Figure 3 foods-14-02048-f003:**
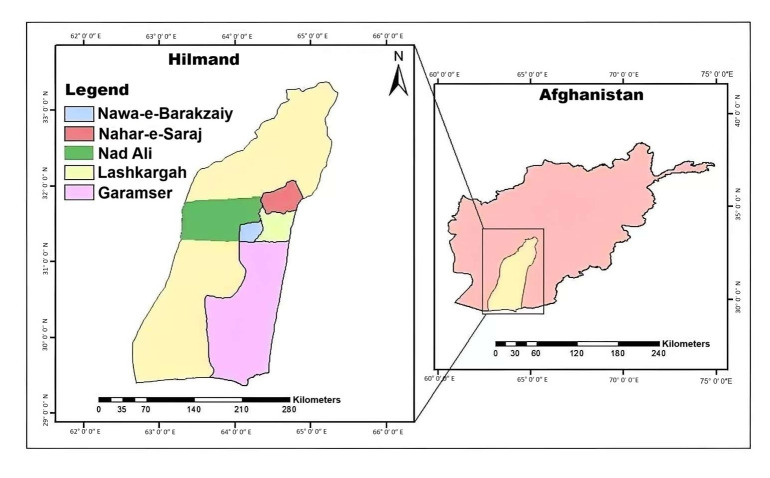
Geographical position of Helmand Province, Afghanistan.

**Table 1 foods-14-02048-t001:** Descriptive statistics of household food insecurity and income diversification in Helmand, Afghanistan. n (302).

Variable	Household Food Insecurity n (302)						
	Food-Secure (5.3%)	Mildly Food-Insecure Access (8.6%)	Moderately Food-Insecure Access (22.5%)	Severely Food-Insecure Access (63.6%)			
	Frequency (Percentage)/Mean (SD)	Frequency (Percentage)/Mean (SD)	Frequency (Percentage)/Mean (SD)	Frequency (Percentage)/Mean (SD)	Total	Pearson’s Chi-Square	Sig.
Income diversification	4.6 (1.9), min = 1, max = 10	2.1 (1.4), min = 1, max = 6	2.3 (1.4), min = 1, max = 6	1.7 (1.6), min = 1, max = 6	302		0.000
Farmer	15 (5.2%)	24 (8.2%)	62 (21.3%)	190 (65.3%)	291	10.4	0.015
Other employment	1 (9.1%)	2 (18.2%)	6 (54.5%)	2 (18.2%)	11		
**Total**	16 (5.3)	26 (8.6%)	68 (22.5%)	192 (63.6%)	302		
Educated	9 (7.3%)	24 (19.5%)	61 (49.6%)	29 (23.6%)	123	150.0	0.000
Illiterate	7 (3.9%)	2 (1.1%)	7 (3.9%)	163 (91.1%)	179		
**Total**	16 (5.3)	26 (8.6)	68 (22.5)	192 (63.6)	302		
Annual income							
Low income	6 (3.1%)	17 (8.8%)	41 (21.2%)	129 (66.8%)	193	31.2	0.000
Middle income	6 (6.1%)	7 (7.1%)	27 (27.3%)	59 (59.6%)	99		
High income	4 (40.0%)	2 (20%)	0 (0.0%)	4 (40.0%)	10		
**Total**	16 (5.3)	26 (8.6)	68 (22.5)	192 (63.6)	302		
Small (10 or less)	15 (7.5%)	22 (10.9%)	56 (27.7%)	108 (53.5%)	201	25.9	0.000
Large (More than 10)	1 (1.0%)	4 (4.0%)	12 (11.9%)	84 (83.2%)	101		
**Total**	16 (5.3%)	26 (8.6%)	68 (22.5%)	192 (63.6%)	302		
Income compared to the previous year							
Less	0 (0.0%)	4 (5.1%)	0 (0.0%)	75 (94.9%)	79	78.3	0.000
Same	3 (3.2%)	1 (1.1%)	24 (25.3%)	67 (70.5%)	95		
More	13 (10.2%)	21 (16.4%)	44 (34.4%)	50 (39.1%)	128		
**Total**	16 (5.3%)	26 (8.6%)	68 (22.5%)	192 (63.6%)	302		
No	0 (0.0%)	0 (0.0%)	7 (4.5%)	149 (95.5%)	156	143.2	0.000
Yes	16 (11%)	26 (17.8%)	61 (41.8%)	43 (29.5%)	146		
**Total**	16 (5.3%)	26 (8.6%)	68 (22.5%)	192 (63.6%)	302		
Rented	0 (0.0%)	1 (10.0%)	1 (10.0%)	8 (80.0%)	10	166.7	0.000
Host with family or relatives	0 (0.0%)	2 (3.0%)	15 (22.4%)	50 (74.6%)	67		
Temporary shelter	0 (0.0%)	1 (0.8%)	8 (6.2%)	121 (93.1%)	130		
Own house	16 (16.8%)	22 (23.2%)	44 (46.3%)	13 (13.7%)	95		
**Total**	16 (5.3%)	26 (8.6%)	68 (22.5%)	192 (63.6%)	302		
Relative holds a government job							
No	1 (0.5%)	2 (0.9%)	32 (14.7%)	182 (83.9%)	217	157.5	0.000
Yes	15 (17.6%)	24 (28.2%)	36 (42.4%)	10 (11.8%)	85		
**Total**	16 (5.3%)	26 (8.6%)	68 (22.5%)	192 (63.6%)	302		
Financial situation							
Comfortable	16 (19.0%)	22 (26.2%)	34 (40.5%)	12 (14.3%)	84	144.4	0.000
Struggling	0 (0.0%)	4 (1.8%)	34 (15.6%)	180 (82.6%)	218		
**Total**	16 (5.3%)	26 (8.6%)	68 (22.5%)	192 (63.6)	302		
Farm size	16.4 (5.1), min = 8, max = 25	9.0 (5.3), min = 1, max = 20	4.8 (3.8), min = 0, max = 20	0.7 (1.6), min = 0, max = 12			0.000
Marital status							
Currently married	16 (6.9%)	24 (10.3%)	8 (3.4%)	185 (79.4%)	233	213.2	0.000
Currently single	0 (0.0%)	2 (2.9%)	60 (87.0%)	7 (10.1%)	69		
**Total**	16 (5.3%)	26 (8.6%)	68 (22.5)	192 (63.6%)	302		
Male family members	9.5 (2.6), min = 5, max = 15	8.3 (2.6), min = 4, max = 16	7.5 (2.9), min = 2, max = 17	5.6 (2.2), min = 1, max = 15	302		0.000
Female family members	5.9 (1.8), min = 4, max = 11	5.0 (2.7), min = 1, max = 12	6.3 (3.2), min = 2, max = 23	9.1 (3.4), min = 2, max = 21	302		0.000
Children under 16	6.6 (2.0), min = 3, max = 11	5.7 (3.4), min = 2, max = 18	6.2 (3.7), min = 2, max = 20	8.6 (3.0), min = 2, max = 19	302		0.000
Distance from DB	13.5 (6.0), min = 8, max = 33	17.8 (13.1), min = 8, max = 54	15.9 (8.5), min = 8, max = 54	24.5 (11.3), min = 8, max = 63	302		0.016
Distance from CT	45.6 (51.5), min = 13, max = 167	66.2 (62.1), min = 13, max = 188	58.3 (38.7), min = 13 Km, max = 188	76.8 (54.0), min = 13, max = 200	302		0.000
Access to transport							
No	4 (2.0%)	14 (7.0%)	43 (21.4%)	140 (69.7%)	201	18.1	0.000
Yes	12 (11.9%)	12 (11.9%)	25 (24.8%)	52 (51.5%)	101		
**Total**	16 (5.3%)	26 (8.6%)	68 (22.5%)	192 (63.6%)	302		

SD: standard deviation, Min: minimum, Max: maximum, DB: district bazaar, CT: capital town.

**Table 2 foods-14-02048-t002:** Household Food Insecurity Access Scale (HFIAS).

Do You or Your Household Members Face the Following Challenges in Maintaining Food Security Because of Financial Problems/Lack of Resources?	Last Month’s Frequency (%)				
	Response (%)		Rarely	Sometimes	Often
Worried about being without food	No	8.28			
	Yes	91.72	38.40	29.80	23.50
You cannot eat whenever you desire	No	11.26			
	Yes	88.74	39.10	25.80	23.80
Decrease the variety and quality of meals	No	11.90			
	Yes	88.10	32.80	32.50	22.80
Consume some meals you did not desire to eat	No	12.90			
	Yes	87.10	31.50	33.10	22.50
Have less food than seems required	No	16.60			
	Yes	83.40	39.40	29.50	14.60
Limit foods eaten in quantities	No	24.50			
	Yes	75.50	32.50	23.20	19.90
There is nothing to eat in the family	No	37.70			
	Yes	62.30	21.20	26.20	14.90
Go to your bed at night hungry	No	45.40			
	Yes	54.60	23.20	19.20	12.30
Go the entire day and night without eating anything	No	41.10			
	Yes	58.90	28.10	20.50	10.30

**Table 3 foods-14-02048-t003:** Household income sources in the past 12 months.

Income Sources	Response	N	Percentage (%)
Contract farming	Yes	175	28.3
Animal products	Yes	120	19.4
Staple grains	Yes	60	9.7
Vegetables	Yes	41	6.6
Self-employment	Yes	41	6.6
Temporary employment	Yes	38	6.1
Handicraft production	Yes	26	4.2
Garden fruits	Yes	23	3.7
Poultry products	Yes	15	2.4
Seasonal migration	Yes	11	1.8
Remittances	Yes	10	1.6
Permanent employment	Yes	10	1.6
Fishery products	Yes	9	1.5
NGO programs	Yes	9	1.5
Receiving gifts	Yes	8	1.3
Small-scale trade	Yes	5	0.8
Government programs	Yes	5	0.8
Lending farmland	Yes	4	0.6
Renting out shop or house	Yes	4	0.6
Vehicle for transport	Yes	2	0.3
Tractor	Yes	1	0.2
Thresher	Yes	1	0.2
Pension	Yes	0	0.0
Total		618	100

**Table 4 foods-14-02048-t004:** Bivariate and multivariate ordered logistic models predicting severe food insecurity.

	Bivariate Odds Ratio	Model 1 (Socio-Economic) Odds Ratio	Model 2 (Bio-Socio-Cultural) Odds Ratio	Model 3 (Locational) Odds Ratio
Income diversification	0.622 (0.08) ***	0.491 (0.15) ***	0.517 (0.16) ***	0.490 (0.17) ***
Main occupation (ref: Other employment)				
Farmer	4.073 (0.56) *	0.630 (0.79)	1.222 (0.84)	0.989 (0.99)
Education (ref: Educated)				
Illiterate	25.509 (0.32) ***	11.830 (0.49) ***	10.867 (0.55) ***	14.813 (0.60) ***
Annual income (ref: High income)				
Middle income	6.57 (0.61) **	0.532 (0.99)	0.220 (1.05)	0.472 (1.13)
Low income	8.76 (0.60) ***	0.560 (0.99)	0.290 (0.99)	0.737 (1.11)
Household size (ref: Large (More than 10))				
Small (10 or less)	0.234 (0.29) ***	0.236 (0.50) **	0.279 (0.53) **	0.283 (0.55) *
Income compared to the previous year (ref: More)				
Same	3.99 (0.28) ***	5.465 (0.52) ***	4.49 (0.56) **	6.40 (0.59) **
Less	28.69 (0.31) ***	2.083 (0.70)	1.89 (0.75)	2.147 (0.79)
Land ownership (ref: Yes)				
No	52.81 (0.43) ***	6.054 (0.74) **	6.475 (0.79) **	6.826 (0.81) **
Residence tenure status (ref: Own house)				
Temporary shelter	84.049 (0.44) ***	2.258 (0.74)	2.034 (0.80)	2.078 (0.84)
Host with family or relatives	18.551 (0.38) ***	2.778 (0.63)	2.257 (0.68)	2.651 (0.72)
Rented	22.832 (0.81) ***	4.117 (1.30)	6.139 (1.42)	7.930 (1.58)
Relative holds a government job (ref: Yes)				
No	43.610 (0.35) ***	4.67 (0.50) **	4.18 (0.56) **	4.733 (0.59) **
Financial situation (ref: Comfortable)				
Struggling	32.714 (0.33) ***	17.85 (0.56) ***	14.639 (0.59) ***	16.181 (0.64) ***
Farm size	0.603 (0.05) ***	0.783 (0.07) ***	0.793 (0.06) ***	0.783 (0.06) ***
Marital status (ref: Currently single)				
Currently married	6.043 (0.28) ***		0.553 (0.49)	0.414 (0.52)
Male family members	0.702 (0.05) ***		0.682 (0.12) ***	0.666 (0.13) ***
Female family members	1.412 (0.05) ***		1.164 (0.10)	1.232 (0.11) *
Children under 16	1.256 (0.04) ***		1.232 (0.12)	1.190 (0.12)
Distance from district bazaar	1.107 (0.02) ***			1.083 (0.03) **
Distance from the capital town	1.008 (0.003) **			1.001 (0.01)
Access to transport (ref: Yes)				
No	2.412 (0.24) ***			0.471 (0.63)
Constant cut1		−7.19 (1.63) ***	−8.67 (1.94) ***	−7.56 (1.99) ***
Constant cut2		−4.40 (1.48) **	−5.69 (1.80) **	−4.43 (1.97) **
Log-likelihood		−96.84	−89.21	−83.42
Goodness-of-Fit				
Pearson’s Chi-Square		357.009 (df = 672), *p* = 1.000	529.725 (df = 884), *p* = 1.000	697.762 (df = 881), *p* = 1.000
Deviance		190.920 (df = 672), *p* = 1.000	178.425 (df = 884), *p* = 1.000	166.858 (df = 881), *p* = 1.000
Pseudo-R-Square				
Cox and Snell R^2^		0.736	0.751	0.760
Nagelkerke R^2^		0.853	0.871	0.882
McFadden R^2^		0.672	0.702	0.721
Observation	302	302	302	302

The dependent variable is HFIAS *** *p* ≤ 0.001, ** *p* ≤ 0.01, * *p* ≤ 0.05. Robust standard errors (in parentheses) are adjusted for clustering to account for intra-group correlation. The log-likelihood value represents how likely it is that the observed data would occur given the model’s parameters. A more negative value indicates a poorer fit. Pearson’s Chi-Square and Deviance Chi-Square tests assess the fit of the model. A high *p*-value (close to 1.0) indicates that the model fits the data well. Pseudo-R-squared measure indicates how much variation in the outcome is explained by the predictors. However, it cannot reach a value of 1. Nagelkerke R^2^ is an adjusted version of Cox and Snell R^2^, allowing it to range from 0 to 1. Higher values suggest a better fit. McFadden R^2^ is Another Pseudo-R-squared measure used for logistic regression. Like other Pseudo-R^2^ measures, it is not directly comparable to the R^2^ of linear regression, but higher values indicate a better model fit.

## Data Availability

The original contributions presented in the study are included in the article, further inquiries can be directed to the corresponding author.

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
