# Peer review of "Household Income Diversification and Food Insecurity: A Case Study of Afghanistan"

_foods, 2025, doi:10.3390/foods14122048_

Round 1

Reviewer 1 Report

Comments and Suggestions for Authors

The issue of food insecurity, particularly in countries like Afghanistan, has high social and human relevance and, as such, deserves scientific attention. This manuscript not only presents the diversity of causes of food insecurity in dialogue with Household income diversification but also presents well-founded suggestions to precisely reduce food insecurity.

From a methodological point of view, the manuscript presents the clarity and rigor required in a scientific article. The discussion of the results is well-founded on the data collected and dialogues consistently with the bibliography. In fact, the extensive and adjusted bibliography used in the text clearly enhances the robustness of the entire argumentative process.

Considering the theme and the importance of social and cultural dimensions, recognized by the authors, it seems to me that it would be valuable for the work to incorporate qualitative methodological strategies, notably with the use of ethnographic practices, as they would allow a contextualization of the phenomenon and privileged access to the discourses and practices of the natives. In the final part of the manuscript this is acknowledged by the authors and forwarded for future work.

Finally, I would just like to point out that, from my point of view, the recurring replacement of the names of the authors cited throughout the text by numbers referring to the end of the text makes reading difficult, as sometimes the subject is a number, which requires a constant movement from the text to the bibliography, resulting in fragmentation of reading and attention.

Author Response

Reviewer #1

  1. From my point of view, the recurring replacement of the names of the authors cited throughout the text by numbers referring to the end of the text makes reading difficult, as sometimes the subject is a number, which requires a constant movement from the text to the bibliography, resulting in fragmentation of reading and attention

Response: Thank you for your valuable comments and helpful suggestions.

Done as requested. We removed the numbers as a subject and cited them at the end of sentences.

Please see “Lines 9-12(p. 22), 21-22(p. 22), 58-59(p. 23), 61-63(p. 23), 68-70(p. 23), 71-72(p. 23), 78-85(p. 23), 89-94(p. 23), 119-120(p. 24), and 122-124(p. 24)”.

Reviewer 2 Report

Comments and Suggestions for Authors

It is necessary to describe in detail the study region in relation to the agricultural commodities produced and the limiting factors that growers face each agricultural season.

A methodological diagram should be included.

The references are too many, they should be limited to what is necessary and updated.

The MS lacks a discussion on externalities such as climate change, food geopolitics and aspects such as access to agricultural inputs.

The MS does not discuss sustainability issues.

Author Response

  1. It is necessary to describe in detail the study region in relation to the agricultural commodities produced and the limiting factors that growers face each agricultural season.

Response: Thank you for your valuable comments and helpful suggestions.

Done as requested.

Please see Lines 313-324(p.8).

  1. A methodological diagram should be included.

Response: Done as requested.

Please see lines 297-301(p.7) and Figure 2(p.7).

  1. The references are too many, they should be limited to what is necessary and updated.

Response: We have reduced the number of references to focus on the most relevant and recent sources. However, since the literature review section has to be developed, as noted by Reviewer 3, additional references were necessary to properly support that development.

  1. The MS lacks a discussion on externalities such as climate change, food geopolitics and aspects such as access to agricultural inputs.

Response: Done as requested. Please see “Lines 44-47(p.2), 84-90 (p.2), 159-166 (p.4), and 203-214 (p.5)”. 

  1. The MS does not discuss sustainability issues.

Response: Done as requested.

Please see Lines 48-52 (p.2), 110-115 (p.3), and 156-158 (p.4).

Reviewer 3 Report

Comments and Suggestions for Authors

Dear Authors,

The manuscript entitled “Household income diversification and food insecurity: A case study of Afghanistan” deals with an important and current topic. It has only minor issues to be handled.

In the Abstract, it is unnecessary to mention the specific odds ratio with p value. The term “keyword” is among keywords. Please, refer to more updated data on the decrease of undernourished persons (the period of 1990–1992 was more than 30 years ago). Literature review has to be developed.

You mention animal husbandry as an income source different from agriculture (lines 168-169), whereas the latter is part of the former. Please, specify the values of p and E you used in Equation (1). More information is needed on the specific sampling process: e.g. how were 302 rural homes randomly selected?; how many units are in one district from which 2 were picked and why two?; how were three villages chosen within each unit and why 3?; why ten to eleven households were questioned from each village and how were they selected? Lines 210-213 are unnecessary repetitions. The last sentence of section 2.4.2 should be the first one of this section. What is the indication of a VIF value of 2.1? The presentation of the sample distribution based on basic background variables (e.g. gender, age) is missing.

When discussing the content of Table 1, please, indicate whether the differences are significant or not (indicate this in the table, too). In the header of Table 1 (n = 302) is the correct form. In Table 3, poultry products make up a separate category, whereas they are part of livestock products. The lack of pensioners in the sample may indicate that the respondents were younger (that is why it is needed to provide information on the age of the respondents).

It is unnecessary to mention “female family members” in line 38, since male family members indicate the other side of the same relationship.

Formatting of in-text citations is not proper, when more than one source is cited. Moreover, there are several inconsistencies in the formatting of the References list. Please, do not refer to a figure as “below”, since its specific place will depend on the final editing of the paper. Empty lines (156-160) are unnecessary.

Comments on the Quality of English Language

There are some typos, grammar and wording mistakes in the manuscript, see, e.g., lines 77, 209, 262, 278, 63 (p. 15), 21 (p. 19), 34 (p. 19), 37 (p. 19), 42 (p. 19), 58 (p. 20), 71 (p. 20), 90 (p. 20). Please, revise the sentences in lines 283-284, 45-47 (p. 19), and 158-161 (p. 22).

Author Response

  1. In the Abstract, it is unnecessary to mention the specific odds ratio with p value.

Response: Thank you for your valuable comments and helpful suggestions.

Done as requested.

  1. The term “keyword” is among keywords (remove).

Response: Done as requested.

  1. Please, refer to more updated data on the decrease of undernourished persons (the period of 1990–1992 was more than 30 years ago).

Response: Done as requested.

Please see Line 38-41(p.1).

  1. Literature review has to be developed.

Response: Thank you for your comments. Done as requested.

Please see the “1.1 Literature review” section.

  1. You mention animal husbandry as an income source different from agriculture (lines 168-169), whereas the latter is part of the former(removed).

Response: Thank you for your comments. removed as requested.  Please see Line 311(p.8).

  1. Please, specify the values of p and E you used in Equation (1).

Response: Thank you for your comments. Done as requested.

  1. More information is needed on the specific sampling process: e.g. how were 302 rural homes randomly selected?; how many units are in one district from which 2 were picked and why two?; how were three villages chosen within each unit and why 3?; why ten to eleven households were questioned from each village and how were they selected?

Response: Done as requested.

Please see line 354-365(p.9).

  1. Lines 210-213 are unnecessary repetitions.

Response: Removed as requested.

  1. The last sentence of section 2.4.2 should be the first one of this section.

Response: Removed as requested. Please see lines 397-399 (p.10).

  1. What is the indication of a VIF value of 2.1?

Response: Thank you for your comments.

Please see line 445-451 (p.11).

  1. The presentation of the sample distribution based on basic background variables (e.g. gender, age) is missing.

Response: In our study, gender was not included in the bivariate and multivariate ordered logistic regression models due to the fact that the data collection in Helmand Province, Afghanistan, only included male participants. This restriction arises from cultural norms and safety concerns that prevent interviews with females in this region.

Regarding the age some other variables, they were excluded from the analysis because they exhibited high standard errors, which may have compromised the stability and interpretability of the regression models. As such, age and some other variables were not considered a reliable predictor in the context of this study.

  1. In the header of Table 1 (n = 302) is the correct form.

Response: Done as requested. 

  1. In Table 3, poultry products make up a separate category, whereas they are part of livestock products.

Response: Thank you for your comments.

We changed it into Animal products

  1. The lack of pensioners in the sample may indicate that the respondents were younger (that is why it is needed to provide information on the age of the respondents).

Response: Thank you for your comments.

The absence of pensioners in the sample is not indicative of a younger demographic, but rather the result of specific socio-political and economic factors in Afghanistan during the period of data collection. The following key points help explain the lack of pensioners:

1.Government Employment and Pension Eligibility: In Afghanistan, pension benefits are typically available to individuals who have worked in governmental positions. However, pension eligibility is limited to those employed in the public sector. Respondents who were employed in the private sector do not receive pension benefits, which explains the lack of pensioners among the sample.

2.Policy Change and Suspension of Pension Issuance: A significant policy change occurred following the recent changes in government leadership in Afghanistan. The new government implemented a policy that effectively suspended the issuance of pensions to retired government employees. This policy shift has had a direct impact on pension distribution, which further explains the absence of pensioners in the sample.

3.Demographic Characteristics: While the absence of pensioners may suggest a lack of older respondents, it is important to note that the respondents in this study were primarily selected from rural households, where the majority of individuals are employed in agriculture or informal sectors rather than government jobs. Additionally, the demographic composition of the sample reflects the broader socio-economic conditions in Helmand Province, where access to formal employment and pension systems is limited.

Therefore, the lack of pensioners is not an indicator of younger respondents, but rather a reflection of the specific political, economic, and employment conditions in Afghanistan during the data collection period.

  1. It is unnecessary to mention “female family members” in line 38, since male family members indicate the other side of the same relationship.

Response: Thank you for your comments. Removed as requested.

Please see line 38 (p.22).

  1. Formatting of in-text citations is not proper, when more than one source is cited. Moreover, there are several inconsistencies in the formatting of the References list. Please,

Response: Thank you for your comments.

As per the recommendation, we have carefully reviewed and corrected the formatting of all in-text citations, ensuring that multiple sources are cited appropriately and consistently according to the journal’s referencing style. Additionally, we have thoroughly revised the References list to eliminate inconsistencies and ensure uniform formatting throughout.

  1. Please, do not refer to a figure as “below”, since its specific place will depend on the final editing of the paper.

Response: Thank you for your comments. We deleted it as requested. 

Please see line 289 (p.6).

  1. Empty lines (156-160) are unnecessary.

Response: Done as requested.

  1. Comments on the Quality of English Language.

There are some typos, grammar and wording mistakes in the manuscript, see, e.g., lines 77, 209, 262, 278, 63 (p. 15), 21 (p. 19), 34 (p. 19), 37 (p. 19), 42 (p. 19), 58 (p. 20), 71 (p. 20), 90 (p. 20).

Response: Thank you for your comments. Done as requested.

Please see lines 76-78 (p.2), 361-363 (p.9), 416 (p.10),432-434(p.10),57(p.19),20-22(p.22), 35-36(p.22),38(p.22),42(p.22),58-59(p.23),71-73(p.23),89-90(p.23).

22 Please, revise the sentences in lines 283-284, 45-47 (p. 19), and 158-161 (p. 22).

Response: Thank you for your comments. Done as requested.

Please see Lines 437-439(p.10 and 11), 44-46(p.22), and 158-161 (p. 25).

Round 2

Reviewer 2 Report

Comments and Suggestions for Authors

The MS is OK